# Publication bias in the social sciences since 1959: Application of a regression discontinuity framework

**Julia Jerke**[1‡]*, **Antonia Velicu**[2], **Fabian Winter**[2], **Heiko Rauhut**[2‡]*

**1** Swiss National Science Foundation, Bern, Switzerland, **2** Department of Sociology, University of Zurich, Zurich, Switzerland

‡ JJ is Lead and first author with the largest contribution. HR is the PI.
* jerke@soziologie.uzh.ch (JJ); rauhut@soziologie.uzh.ch (HR)

**Data Availability Statement:** Data and code for the reproduction of our results can be found at https://doi.org/10.17605/OSF.IO/J58CF.

## Abstract

While publication bias has been widely documented in the social sciences, it is unclear whether the problem aggravated over the last decades due to an increasing pressure to publish. We provide an in-depth analysis of publication bias over time by creating a unique data set, consisting of 12340 test statistics extracted from 571 papers published in 1959-2018 in the Quarterly Journal of Economics. We, further, develop a new methodology to test for discontinuities at the thresholds of significance. Our findings reveal, that, first, in contrast to our expectations, publication bias was already present many decades ago, but that, second, bias patterns notably changed over time. As such, we observe a transition from bias at the 10 percent to bias at the 5 percent significance level. We conclude that these changes are influenced by increasing computational possibilities as well as changes in the acceptance rates of scientific top journals.

## 1 Introduction

The increasing *marketization of science* created a breeding ground for biases in the scientific literature [1]. Due to the introduction of success criteria such as the journal impact factor and h-index to measure academic performance as well as the increasingly exclusive acceptance rates for manuscripts in scientific top journals, incentives to follow individual self-interest have increased nowadays, leading to a situation where researchers face a highly competitive 'publish or perish' system. Successful careers increasingly require constant production of original, novel and innovative contributions, which are expected to be published in highly ranked journals and become highly cited [2–4]. On the one hand, more pressure is likely to prompt researchers to work on more selective, more surprising results, which are also more likely to represent selective outliers or biased or even tweaked findings. On the other hand, journals aim to establish impact and reputation by publishing papers that are likely to receive a high number of citations. Given the fact that a paper's quality is often confused with the statistical significance of the reported results, this may lead editors and reviewers, and to prefer hypothesis-confirming results [5, 6]. Closely related to this is a phenomenon called "winner's curse" in

**Funding:** Swiss National Science Foundation; Starting Grant CONCISE, BSSGI0-155981.

auction theory [7]. New significant results on the most original hypotheses win the race of getting access to prestigious journals, but at the same time overestimate the strength of significant effects. Strong evidence for this thesis is that replications, even when they can replicate an original finding, systematically find weaker effects than those reported in the original study [8]. As a result, the scientific literature is biased towards overly positive findings, resulting in an unnatural over-representation of significant findings—one form of manifestation of the so-called publication bias.

Publication bias can be described as a bias in favor of predominantly publishing significant, hypothesis-conforming results. It has been shown that manuscripts reporting positive or significant results have a higher publication probability, independently from the methodological quality [9]. In particular this can imply that a paper reporting significant estimates is published, whereas a paper of equal theoretical, conceptual and methodological quality with insignificant results is rejected or not even submitted. Despite a wide range of definitions and perspectives formulated by different authors, they largely agree that the published literature may not be representative for the universe of performed research, creating a non-representative accumulation of significant results in the published state of the art.

The presence of a substantial publication bias has been clearly documented in many disciplines, including the social sciences [10–14]. Most of the evidence is indirect. For example, funnel plots in meta-analyses provide information on variance and mean values of effects as a function of standard error. In a remarkable meta-analysis Doucouliagos and Stanley [15] constructed a funnel plot for 1424 studies on the effects of minimum wage laws on unemployment. Rarely there is an opportunity for direct observations of publications bias. An exception are the studies by Turner (et al. [16]) and Franco (et al. [13]) who managed to look directly into the file drawer of pharmacological studies and sociological studies, respectively. Because the studies were preregistered, the published and non-published results could be compared. It turned out that almost all significant results were published, while non-significant results remained in the file drawer.

Even though the problem of publication bias has received increasing attention–evident, for example, in the growing number of studies dealing with it–it remains unclear whether publication bias is actually a new phenomenon or whether it already existed for some time. Overall, the evidence in this respect is mixed. Several studies indicate that the scientific literature suffered from a substantial over-representation of significant results already several decades in the past [10, 17–19]. More recent studies, however, suggest that the bias might have increased over time [20, 21]. Other studies, in contrast, indicate that bias patterns did not change throughout the last years [22], or that the bias even decreased recently [23].

We contribute to this gap in the literature by presenting the longest and most fine-grained data set on publication bias in the social sciences, more specifically in the economic discipline. Our sample comprises all studies published between 1959 and 2018 in the *Quarterly Journal of Economics*, and therefore guarantees a complete account of the last sixty years. To investigate publication bias, we focused on quantitative contributions and, first, identified all articles that reported an empirical study in which one or more hypotheses were tested. In a second step we extracted statistical test results from all eligible articles. Overall, we collected 12340 test statistics (*z* or *t*) from 571 papers published between 1959 and 2018. We will examine the empirical distribution of the test statistics for discontinuities at the common levels of significance and whether the patterns change over time.

Our study thereby further contributes to the methodological literature. We propose a novel, more fine-grained and efficient procedure for detecting publication bias compared to previous studies. In general, we test whether we find clustering of significant results at the right-hand side of the common significance levels in the empirical distribution of test statistics. We,

therefore, make use of the principle of regression discontinuity for our specific case. We apply a procedure proposed by McCrary [24] and adapt it to testing for discontinuities in the distribution of test statistics. More precisely, we investigate whether the density of results just above the thresholds of significance significantly exceeds the density just below these cutoffs by comparing the left-hand and right-hand limit of a weighted regression at the threshold. In this way, our proposed method allows a more fine-grained picture of publication bias.

In the remainder, we proceed as follows: In the next section, we provide an in-depth analysis of how the cult-like fixation on statistical significance has led to misconceptions and eventually widespread biases in the scientific literature. We further review previous evidence on changes in publication bias over time. We describe our sampling procedure and prominent sample characteristics in the third section. The fourth section is dedicated to deriving our methodology to detect discontinuities in the empirical distribution of test statistics. We present our cross-sectional and longitudinal results on publication bias in section five and six. Eventually, we conclude our investigation with a broad discussion of our findings and concluding remarks regarding policy changes that have the potential to mitigate publication bias in the future.

## 2 Statistical significance, misconceptions and publication bias

### 2.1 How statistical significance testing became a trigger for publication bias

The concept of statistical significance is often misunderstood and mistakenly regarded as an indicator of whether a theory or hypothesis is *true* or *false* [25]. However, not being able to reject the null hypothesis does not imply its confirmation and it particularly does not justify any conclusions about the alternative hypothesis. The same applies if the null hypothesis is rejected; it does not follow that the alternative hypothesis is *true*. This misconception is joined by another widespread but wrong belief: results that do not confirm the alternative hypothesis are less valuable for scientific progress as they do not establish new "truths". Several studies indeed show that scientists often have difficulties with the correct interpretation of *p* values and statistical significance [26–28]. Further, an analysis of articles published in the European Sciological Review indicates that a substantial proportion of published research is prone to a misuse of statistical null hypothesis testing [29]. For instance, in roughly half of the articles non-significant effects are interpreted as zero effects. In summary, statistical significance often seems to be erroneously confused with importance, and significance testing too often boils down to a binary decision about the existence of an effect instead of discussing its magnitude [30].

Criticism against the concept of statistical significance has already been voiced just after it gained currency. For instance, [31, p. 474] stated: "[. . .] tests of significance, when used accurately, are capable of rejecting or invalidating hypotheses, in so far as these are contradicted by the data; but [. . .] they are never capable of establishing them as certainly true. In fact [. . .] 'errors of the second kind' are committed only by those who misunderstand the nature and application of tests of significance." Selvin [32] even warned of the possible threat of creating incentives to manipulate statistical analyses. Nevertheless, tests of statistical significance became broadly accepted and are now one of the most commonly applied standards for judging empirical research.

The meanwhile heavy use of significance testing quite clearly contributed to the phenomenon of publication bias and other far-reaching, unexpected and presumably unintended consequences for what is published in science. For instance, Ioannidis (et al. [33]) re-examined 159 meta-analyses from various economic areas and demonstrate that the statistical power of the analyses generally did not exceed 18 percent and that 80 percent of the meta-effects are

overestimated by a factor of at least two. They argue that underpowered studies with significant findings are often merely reporting random noise or bias [34, 35]. This is also reflected in the overall proportion of significant findings in the scientific literature. Using a simple Bayesian argument, Ioannidis [34] shows that the conditional probability that effects reported as significant are really true can be very small. Under certain conditions, in particular a low a priori probability of the truth of the hypothesis, a low statistical power, and the convention of $\alpha$ = 5%, it follows that "most published research findings are false" [34]. The majority of published findings is significant; a pattern that did not change much over the last decades and that can hardly be attributed to testing only plausible hypotheses [3, 10, 20, 36, 37]. In a comparison of publications from different disciplines, Fanelli [3] shows that the range of "positive" results varies from space science to psychology from about 70% to more than 90%. Thus, we can safely assume an under-representation of non-significant results as a consequence.

## 2.2 Did increasing publication pressures lead to more publication bias? Review and problems of previous trend studies

Gerber and Malhotra [9, p. 11] note that "evidence of publication bias has been found throughout the social sciences since the 1950s". Indeed, there are studies on publication bias which date back several decades ago. One of the first scientists drawing attention to a potentially biased literature is Sterling [17]. In his study, he found that out of a sample of roughly 300 psychological articles more than 97% reported significant results. Some thirty years, later Sterling [37] repeated the study and found comparable patterns, concluding that there were no changes in publication bias over the years. This conclusion also implies that the tendency towards predominantly significant results already existed in the Fifties. A similar over-representation of significant results was also found in economics [10], in sociology [18] and medicine [19].

Despite the fact that there were already signs of publication bias in earlier years, we hypothesize that the bias has increased notably over the past decades. Two major arguments support this notion. First, academia has experienced an enormous increase in competition. Due to declining acceptance rates for manuscripts in top journals and the increasing measurement of academic success by hard indicators, such as the journal impact factor and h-index, scientists and journals are now under more pressure than ever [38, 39]. This might have resulted in ever-more original, innovative but presumably also more selective research being submitted to and selected by journals. Second, with the advent and advancement of statistical software, statistical analysis has become much more efficient and can nowadays also be increasingly automated. From a technological and computational perspective, it has therefore become much easier to repeat or tweak analyses until the output matches the desired results [21], often referred to as p-hacking.

The results from empirical studies explicitly investigating the development of publication bias over time are rather mixed. Fanelli [20], for instance, shows that the proportion of significant findings increased remarkably between 1990 and 2007. He analyzes the results of over 4600 papers published in this period, and finds that across various disciplines (including the social sciences) the average prevalence of positive results has increased by a factor of 1.22 from roughly 70% to about 86%. Further indicative evidence, that publication bias has increased, can, for example, be found in the study of Leggett (et al. [21]), who examine two top psychology journals and find a much higher peak at just below $p$ = 0.05 in the year 2005 than in 1965. In contrast, Brodeur (et al. [22]) find no change in bias patterns in empirical economics over the last fifteen years, and Vivalt [23] even find a decreasing bias for economic studies reporting randomized controlled trials over the last twenty years.

However, the preceding investigations and comparisons of publication bias over time entail two problems and can so far only serve as weak evidence for changes in publication bias over time. First, even though there are studies on publication bias since the 1950s, a comparison with recent investigations only constitutes a comparison of a few single cross-sectional studies in different scientific disciplines. Second, methods to detect publication bias have been largely refined. Over the last ten years new strategies were developed and applied. Thus, comparing older with more recent studies implies comparing results obtained with different methods and from uncomparable data from different disciplines. Therefore, the current state of the art in speculating about time trends of publication bias has to be treated very carefully.

Our investigation addresses both objections and can be regarded as the first study of a long, consistent and fine-grained time trend in the social sciences. We, first, collected all data within one single scientific journal for a time span of sixty years and, thus, created a consistent data set that allows for inference on time effects. Second, we have carefully hand-selected only coefficients which were related to hypotheses to reduce bias as much as possible. Lastly, we develop a new method to test for publication bias and apply it to our unique data structure. This large-scale investigation allows us to directly investigate whether and how bias patterns have changed over time.

## 3 Setting and data

### 3.1 Journal selection

The setting of our empirical work are the social sciences, and more specifically the economic discipline. This discipline is particularly interesting since it encompasses a broad variety of topics and broadly overlaps with other disciplines such as sociology, political science, psychology, education research or health science. The basis of our analyses is the *Quarterly Journal of Economics (QJE)*. We chose the *QJE* because it belongs to the elite group of highly ranked top journals in economics. We focus on the group of top journals as we expect them to pioneer general developments in the discipline. Further, they may often be the first target journals to which manuscripts are submitted and therefore serve as role models of how and what to publish in one discipline. For example, rejected manuscripts which would have fit the *QJE* and are rejected due to the low acceptance rate are often submitted elsewhere, making manuscripts in *QJE* representative of a larger set of top manuscripts in the field. Further, we conjecture that the top journals in a field have a large influence on methodological conventions, working styles and topics in a field; hence potentially biased publications there will have a large influence on topics and methods of upcoming research.

For the sake of conducting an extensive longitudinal analysis, it is not feasible to examine all top journals, forcing us to choose one single journal. Founded in 1886, the *QJE* is the oldest economics journal, supplying us with a simple decision rule that we expect to be uncorrelated with the features we aim to investigate. Meaning, we do not expect to observe patterns in the *QJE* that we would otherwise not see in other top journals of the social sciences.

### 3.2 Sample construction

We will apply a narrow interpretation of publication bias and investigate whether a artificial accumulation of statistical results just above the common thresholds of significance can be observed. We will therefore analyze the empirical distribution of test statistics published in the *QJE*. We focus on the volumes 73 to 133, yielding an observation period that spans six decades, 1959 to 2018. In line with the outlined purpose of our investigation, we construct two data sets. (1) The first data set contains all articles published in the observation period and the respective article meta information (henceforth called *article data set*). (2) The second data set

**Table 1. Selection procedure to construct the *coefficient data set*.**

| A. First selection step | | |
| --- | --- | --- |
| | | N |
| Initial sample: | All publications | 2908 |
| (number of observations in the *article data set*) | | |
| Inclusion criteria | | |
| First stage: | Articles that report empirical studies | 1094 |
| Second stage: | Articles that state and test hypotheses | 775 |
| Third stage: | Articles that report the respective test statistics or standard errors | 609 |
| B. Second selection step | | |
| Coefficients extracted from third stage articles | | 13736 |
| Coefficients corresponding to two-sided tests | | 13451 |
| Coefficients with $abs(z)<10$ | | 12340 |

contains the statistical test results extracted from eligible articles (henceforth called *coefficient data set*). The coding process took four years. Both data sets were manually compiled in a detailed, effortful and standardized procedure and represent a full sample of the literature published in the *QJE* between 1959 and 2018. The main part of the work was due to selecting only those coefficients for which hypotheses were formulated, making it necessary that an expert reads all articles.

**Article data set.** We used the Web of Science Core Collection (WoS) data base to construct an initial sample. Overall, the *QJE* published 2908 items from 1959 to 2018, including notes, comments and replies. The WoS provides basic information such as title, name of author(s), publication year, volume and issue, number of cited references and number of citations since publication. Via thorough content analysis we categorized all articles in either empirical study, theoretical contribution, or discussion (note, comment, reply, etc.).

**Coefficient data set.** After having constructed the *article data set*, we extracted test results following an elaborate two-step selection procedure: First, the selection of eligible articles, and, second, the selection of therein reported effects. To be eligible, articles had to meet three inclusion criteria, which we refer to as the three stages of our first-step selection (see Table 1, panel A).

In the first stage, we explicitly filter articles that report empirical studies and contain appropriate statistical tests. The reason is that we refer to publication bias as a systematic noise induced by preferential selection of significant findings or manipulation of non-significant effects that eventually may result in suspicious patterns in the distribution of published statistical test results. Thus, our operationalization naturally entails the exclusion of theoretical contributions and discussion articles. However, this does not necessarily imply that non-empirical research that develops models or theories is not prone to certain forms of bias. For instance, Paldam [40] refers to *t-hacking* as a form of bias introduced by authors that carefully tailor theoretical assumptions to end with the desired and well publishable conclusion. Certainly interesting, however, this form of bias is not addressed in this study. In some cases, scientific notes also reported empirical studies. Since the majority of these empirical notes directly referred to an article previously published in the *QJE*, it is likely to assume a higher publication probability. Hence, these notes were excluded from the publication bias analysis. However, we included the few cases in which such empirical notes did not refer to articles previously published in *QJE*. Overall, 1094 articles reported empirical studies.

In the second selection stage, we excluded articles that do not report and test hypotheses in the sense that the authors formulate one or more initial expectations regarding the effect or

relationship they investigate. By this means, we dismiss purely descriptive or explorative articles. In line with other studies e.g., [14], we strike a balance between the very restrictive procedure of authors such as Gerber and Malhotra [9], that only include articles stating explicit hypotheses, and authors such as Masicampo and Lalande [41] that do not restrict themselves on articles with hypotheses at all and extract any reported test results. Instead we focus on articles that state either explicit or implicit hypotheses. By this means, we did not only include articles where the central theoretical assumptions were highlighted with keywords such as "hypothesis", "prediction", "proposition" or "expectation" but also articles missing such keywords where the expected effect could be deduced from the text. This made it necessary to read all articles in detail from one expert, making this a very time-consuming enterprise leading to a unique data set. In more detail, we first ran a text search with the keywords "hypoth*", "predict*", "propos*" and "expect*" (abbreviations were used to not erroneously ignore divergent wordings, i.e. "hypothesis" versus "hypothesize") to identify explicit hypotheses. Second, we also carefully reviewed articles not containing any of these keywords for implicit hypotheses. In total 775 articles met the criteria from the first two stages.

In the third and last stage we further excluded articles that failed to provide test statistics or related estimation results that allow their calculation such as standard errors, sample size or $p$ values. Additionally, we also excluded papers that were based on simulated data and papers that reported non-parametric tests only (such as chi-square tests), hence, restricting our data to analyses for which parameters were tested using the Student's t or the normal distribution.

Overall, the above described three-stage selection procedure reduced the initial 2908 publications to 609 eligible articles. These remaining articles all include empirical studies with hypotheses and appropriate test statistics. After this *first-step selection*, we performed the *second-step selection*. The second step describes the selection of extracted test results from the set of eligible articles in order to construct our coefficient data set (see Table 1, panel B).

The main challenge for selecting appropriate coefficients from the remaining eligible articles is the identification of coefficients which relate to hypotheses. Therefore, we carefully read the respective results section and inspected regression tables to clearly link the reported coefficients to the hypotheses. While we have applied broad inclusion criteria of what counts as article with hypotheses, we have been relatively restrictive in deciding which coefficients relate to the stated hypotheses.

In particular, we were careful not to extract multiple coefficients from regression models building upon each other since these are expected to be highly correlated. This procedure prevents potential inflation of our results on publication bias. As a rule of thumb we always extracted the coefficients presented in the full model including control variables. Sole exception were coefficients reported in partial models to which the authors explicitly referred to as evidence for or against their hypotheses. We did not extract test results that were of one of the following characteristics: the coefficient is the result of testing identification assumptions (including results from first stage regressions); the coefficient is linked to a hypothesis expecting a null effect (including randomization checks, placebo tests, falsification checks etc.), or the coefficient is estimated as part of robustness tests related to prior central results. We excluded test coefficients that were expected to be zero since the term publication bias refers to an over-representation of significant results. Coefficients with an expected null-effect on the contrary potentially trigger different mechanisms and strategies that contrast with strategies for coefficients that are expected to be significantly non-zero. Also, for the purpose of parsimony, we excluded robustness tests as they generally repeat previous central analyses applying divergent specifications and, therefore, yielding results that are often similar to those from earlier analyses. Including them would possibly inflate our results. All our analyses we have considered only coefficients corresponding to two-sided tests. In our analyses we focus exclusively

on coefficients we found for two sided tests and that reported z-values smaller than 10 as extremely large values are excessively influential in our test procedure. In total we collected 12340 test results from 571 articles.

We constructed our coefficients data set using *t* values. Note, however, that in most cases these were not reported in the paper. We therefore calculated them by dividing the regression coefficient by its standard error. In a few other cases the authors only reported the coefficient and its *p* value. Using information on the degree of freedom and the respective distribution function we could calculate the test statistic also in these cases.

**Transformation to z-scores and weighing.** We transformed all *t* scores to *z* scores. This was necessary because the majority of our extracted test results stem from regression analyses with different sample sizes, and, therefore, contains *t* statistics with varying corresponding degrees of freedom. In most cases the authors did not report the degrees of freedom. We then calculated it by *df = N − #coefficients in the regression equation*, including the constant and possible fixed effects. Since the Student's *t* distribution depends on the degree of freedom, our *t* statistics will be tested against different theoretical distributions yielding a variety of corresponding critical cutoff values for significance and making it impossible to investigate whether there is bias at the common levels of significance. For each *t* statistic we, therefore, determine the individual *p* value and re-calculate the respective *z* statistic that would get assigned the same *p* value. To be more concrete, we undertake the following transformation:

$$p(t_{ij}) = 2(1 - T.dist[t_{ij}, df_{ij}]) \quad \Rightarrow \quad z_{ij} = Norm.Inv\left[1 - \frac{p(t_{ij})}{2}\right], \tag{1}$$

where $t_{ij}$ is the *t* statistic from coefficient *i* of article *j* and $df_{ij}$ the respective degrees of freedom. $p(t_{ij})$ is the corresponding *p* value and $z_{ij}$ the *z* statistic resulting from the transformation. Following this procedure we receive a consistent distribution of *z* statistics; each one corresponding to a *t* statistic from our data set adjusted for the respective degrees of freedom.

We weigh our data based on the number of coefficients in the respective article to equally account for articles reporting few and many coefficients. We perform this transformation based on the arguments of the preceding study by [14]. We construct the weighting variable using the inverse of the number of coefficients extracted from each article. Hence, the more coefficients from one article the lower their weight. We will report results for raw and imputed data as well as for unweighted and weighted data.

## 3.3 From theory to empirics and towards more complex analysis

Prior to the analyses on publication bias, we will document several important characteristics of the data that might already bear implications with respect to publication bias. Fig 1 shows how the ratio of theoretical to empirical contributions changed over time.

We further observe a rise in complexity of empirical analyses indicated by an increasing number of coefficients that we collected from the articles and by increasing sample sizes. In the first decade (1959–1968) we extracted on average 12.4 coefficients per article and the number of coefficients rises to 29.4 in the most recent decade (2009–2018). Remember that we only extracted core coefficients that were relevant for the decision for or against the hypothesis of interest. It is therefore plausible to attribute the risen number of coefficients per article to more complex and comprehensive analyses. In tandem with the increase in core coefficients per paper, sample sizes have notably increased. Fig 2 shows the distributions of the sample sizes. The left figure in Fig 2 shows the raw distribution of sample sizes over time. We log-transformed the sample size to visually account for very large sample sizes. The roughly linear increase of the logarithmized sample size suggests an exponential increase which is further

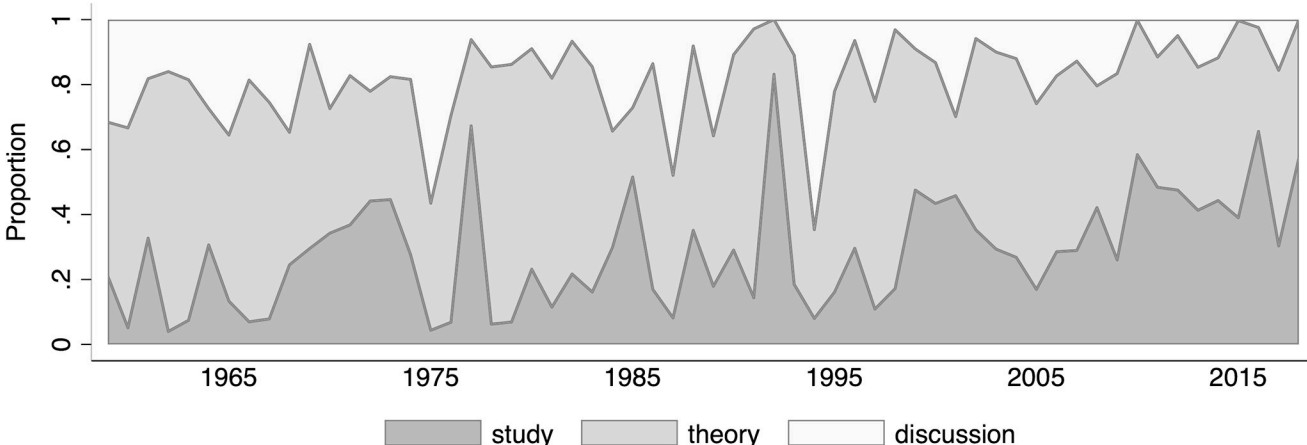

**Fig 1. Composition of the *QJE* over time, 1959–2018.** The graph shows the proportional distribution of theoretical and empirical contributions and discussions (e.g. notes, comments, replies) over the observational period. Shown are stacked proportions.

underlined by the median of sample sizes shown in the right figure (in Fig 2). While the median sample size is 52 before 1990, it increases to 1904 after 1990. The trend towards more complex analyses with larger samples is very likely driven by technological and computational progress and by increasing access to large public data files, but in addition may also be an indication that the expectations from manuscripts eligible for top journals has risen, contributing to the notion of an increasing pressure to publish or perish. The positive side of this development is that with the increasing sample size the power of statistical tests has increased.

Despite the increase in complexity of analyses and sample sizes, we do not see an increase in the share of significant results (see S1 Fig in the S1 File). Overall, 64% of the coefficients are significant at least at the 10 percent significance level. The 5 percent significance level was met by 56% of the coefficients and the 1 percent significance level was met by 41% of the

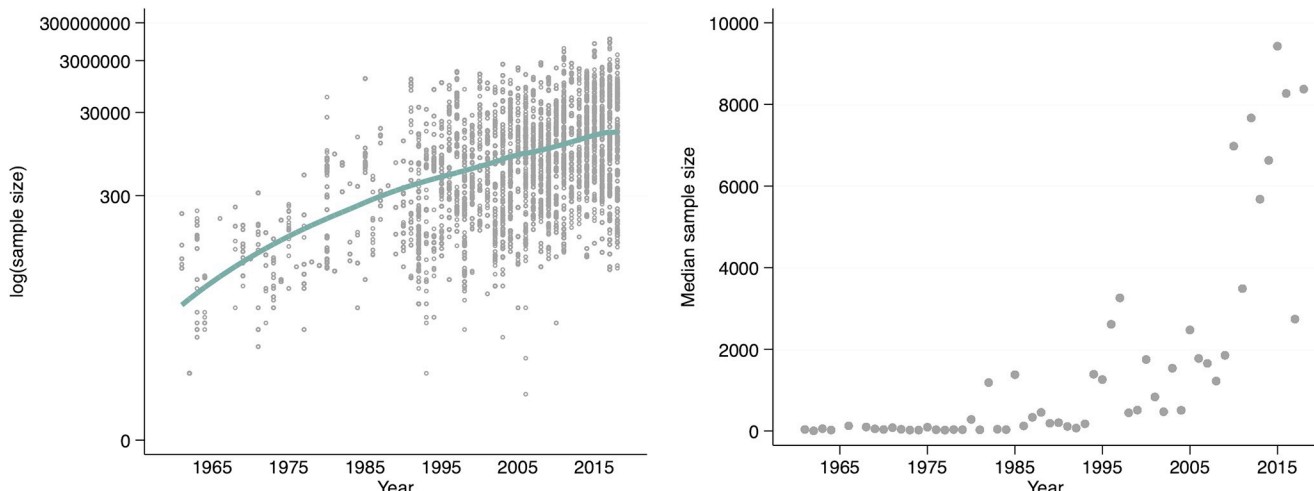

**Fig 2. Sample sizes over time.** Figure on the left shows sample sizes over time. Each dot represents the respective sample size of a single coefficient of our *coefficient sample*. The sample size was log-transformed to account for very large sample sizes. The line indicates a locally weighted regression. Figure on the right shows the median sample sizes for each year.

coefficients. Only 36% of the coefficients in our *coefficient data set* are not significant at any conventional significance level. The lack of an increase in significant findings over time is surprising for two reasons. First, such an increase has been previously documented in other studies e.g., [20]. Second, since the sample size is directly related to the statistical significance of a test statistic, we would have expected the substantial increase in sample size to be associated with increasingly significant results. We performed a robustness check by repeating the visual inspection with the weighted data as well as including the number of coefficients per article in the regression (see S2 Fig and the corresponding regressions in S1 Table in the S1 File), and again do not see an increase over time.

## 4 A novel method for detecting publication bias

### 4.1 Challenges in the measurement of publication bias

There does not yet exist a standard procedure to investigate publication bias. This reflects the general difficulty of measuring the absence of negative results based solely on the body of published results. Most current detection methods of publication bias within samples of heterogeneous effects draw on the empirical distribution of test statistics or *p* values. They share the key assumption that this distribution is supposed to be continuous [21, 22, 41]. Even though the exact shape of the underlying distribution is unknown, it certainly should not have any jumps or discontinuities; and in particular not at the (virtually arbitrary) common critical thresholds of significance such as 1.64 or 1.96 for the ten percent or five percent level of the standard normal distribution.

One frequently used method to investigate publication bias in samples of highly heterogeneous effects is the caliper test [9, 12]. It has meanwhile been applied in various disciplines (e.g., sociology: [12, 42, 43]; economics: [23, 44]; political sciences: [9]). The method is particularly appealing since it is easy to apply and requires only few prerequisites. In brief, the test compares the absolute frequency of test values in narrow and equal-sized intervals just above and just below the critical thresholds of significance and evaluates the difference between those two frequencies with a simple binomial test. However, the test has several weaknesses: 1) it is only based on a very small subset of the distribution of test statistics, 2) the results strongly depend on the size of the intervals and it remains unclear how to determine the optimal width, and 3) the test is not immune to low precision of the reported test statistics and clustering that is independent from publication bias.

Since then, further methods have been proposed to overcome these weaknesses. For instance, [44] extend the basic caliper test with Monte Carlo simulations to account for low precision of reported test results. [21, 41] fit an exponential function to the empirical distribution of *p* values and conduct residual analyses at the critical threshold of significance. [45] compare published test statistics with critical values that are adjusted for a file drawer bias. And [14, 22] construct a counterfactual distribution of test statistics to estimate the inflation of significant results.

Following, we propose a novel method to measure publications bias that also overcomes the weaknesses of the caliper test and combines some of the strengths of the above mentioned advancements. Our approach is based on the principle of regression discontinuity and has several advantageous features that we will briefly discuss. First, we use the full range of the distribution of test statistics, giving higher weight to the area near the significance thresholds. Thus, we do not discard any region of the distribution while still placing a particular focus on the region where publication bias is likely to become visible. Second, the procedure makes use of local linear smoothing and therefore—at least to some extent—corrects for clustering due to low rounding precision. For instance, as [14, 44] point out, a natural clustering at the value

$z = 2$ must be expected since $z$ values equal the ratio of a coefficient and its standard error and since those two values are often reported with a low rounding precision. Thus, by applying a smoothing process, we make sure that a discontinuity at $z = 1.96$ not just reflects a clustering at $z = 2$. Third and last, the procedure is very simple to implement and does not require specific prerequisites.

In what follows, we will describe in detail how we set up our test procedure and explain why it is well suited to investigate publication bias. For the sake of completeness, we will later also rerun our analyses with one of the most commonly used methods to test for publication bias, the caliper test, and report the results in the S1 File.

## 4.2 Regression discontinuity applied to publication bias

We apply a procedure known from the literature on regression discontinuity. Widely used in the social sciences, regression discontinuity is a quasi-experimental design that is usually applied to estimate treatment effects whenever individuals are assigned to a treatment based on an underlying random assignment variable $R$ [46]. Meaning, those whose value for $R$ exceeds a certain cutoff value receive the treatment, the others do not. The treatment effect can then be estimated by comparing the outcome of interest for individuals just below and just above the cutoff assuming that they are virtually similar regarding certain basic characteristics as they only marginally differ in $R$ [47].

Comparable to our problem of detecting unnatural jumps in the distribution at certain cut-offs, regression discontinuity requires that there is no unnatural over-representation of cases just above the cutoff value and, accordingly, also no underrepresentation of cases just below the cutoff. This would otherwise pose a major threat to the validity of regression discontinuity since such jumps in the distribution may indicate manipulation of the assignment variable $R$ by the observed individuals. For example, if individuals anticipate a potential benefit associated with receiving the treatment they may possibly manipulate their value of $R$ to get assigned. The estimated treatment effect will then be biased as the required comparability of individuals near either side of the cutoff is not met. This is also called sorting; meaning that individuals that otherwise would not have been assigned to the treatment now receive the treatment, resulting in a discontinuity in the density distribution of the assignment variable at the cutoff.

[24] introduced an easily applicable procedure that tests for these discontinuities at given cutoff values. We apply this idea to the framework of publication bias. In our case, we do not analyze individuals receiving the treatment, e.g. a social policy benefit once they score higher than a given cutoff, but rather coefficients that receive the treatment *significance* once their corresponding test statistic exceeds the cutoff value of the applied decision rule of statistical significance. For instance, a $z$ statistic exceeding 1.96 (1.64) is considered *significant* on the five (ten) percent level, a $z$ statistic below those values is considered *insignificant*. Note that we actually do not conduct a full-fledged regression discontinuity analysis as we are not interested in an outcome variable after treatment; but we test for discontinuities in the distribution of test statistics by applying the McCrary test.

The McCrary test can be applied in situations where manipulation in $R$ by the individuals is in principle possible and monodirectional [24]. Both is given in our case of application. Manipulation in the distribution of published test results is possible, for instance, by preferential publication of positive and significant effects, driven by journals and authors. The same applies for tweaking of data by researchers to yield significant results. In addition, the manipulation is monodirectional, since the opposite implies that test statistics are manipulated downwards to be insignificant. Conversely, in cases where unrelatedness between two concepts is the desired outcome there may be reasons to manipulate or select coefficients towards non-

significance. But, as explained before, we did not extract coefficients for which a zero effect was predicted. Hence, there is no bias to be expected from that side.

## 4.3 Setup for testing discontinuities at the thresholds of significance

The setup for our application of the McCrary test is as follows ([24] provides further mathematical details). The assignment variable $R$ is the test statistic. The cutoff value $c$, where a discontinuity is expected, is the significance threshold, for instance $c = 1.96$.

The test requires two steps. In the *first* step, an undersmoothed histogram showing the plain density distribution is constructed for $R$. In our case this is simply the empirical distribution of test statistics. In the *second* step, a weighted local linear regression smoothes the histogram by regressing the normalized height of the bins is regressed on the respective bin midpoints. In brief, this means that the height of a particular bin is estimated with a weighted linear regression where the bins closest to the point of interest receive most weight. Local smoothing thereby takes place within the range of a cautiously chosen bandwidth $h$. Continuity in $c$ is then given when the left-hand and the right-hand limits for $c$ are equal and the discontinuity estimate of interest $\theta$ is expressed as the log difference in height:

$$\theta = \ln \lim_{r \searrow c} f(r) - \ln \lim_{r \nearrow c} f(r). \tag{2}$$

where $r$ is the value that the running variable $R$ takes and $f(r)$ is the respective estimation result of the weighted linear regression at $r$.

As McCrary [24] outlines, however, it is more accurate to perform local linear smoothing separately for the left-hand side and the right-hand side of $c$ yielding two separate regressions that can be compared in $c$:

$$\hat{\theta} \equiv \ln \hat{f}^+ - \ln \hat{f}^-. \tag{3}$$

whereas $\hat{f}^+$ is the estimate from the right-hand and $\hat{f}^-$ the estimate from the left-hand regression. It can be shown that $\hat{\theta}$ is asymptotically normal and consistent, and the respective equation for the approximate standard error $\hat{\sigma}_\theta$ to compute confidence intervals is given by

$$\hat{\sigma}_\theta = \sqrt{\frac{1}{nh}\frac{25}{4}\left(\frac{1}{\hat{f}^+} + \frac{1}{\hat{f}^-}\right)}. \tag{4}$$

where $n$ is the sample size and $h$ refers to the chosen bandwidth.

For the test to produce valid results, a suitable bandwidth $h$ for the local smoothing must be chosen such that both excessive noise is avoided when $h$ is very small, and oversmoothing at the cutoff is avoided when $h$ is too large. The McCrary test is implemented in various statistical programs such as Stata and R, and with it comes an automatic procedure to select the appropriate bandwidth. In the following analyses we refer to this bandwidth as the *default bandwidth*. The automatic bandwidth selection is based on a two-step procedure. First, a histogram with bin size $b = 2\hat{\sigma}n^{-\frac{1}{2}}$ is created where $\hat{\sigma}$ is the standard deviation of the sample of test statistics. Second, to compare the observed binned distribution to the distribution one would expect without interference, a fourth-order polynomial regression is estimated on the right-hand and the left-hand side of the cutoff. The optimal bandwidth is then based on a bandwidth selector [48] and is given as the average of a left-hand and right-hand estimated expression that is featuring the mean-squared error of the polynomial regression, its second derivative and a constant derived from the integrals of the kernel. See [24, p. 705] for the exact calculation formula. To investigate the sensitivity of our results we will additionally provide results with varying

bandwidths and determine the optimal bandwidth by mainly relying on visual inspection of the discontinuity plots that we provide together with the quantitative estimation results. The McCrary procedure not only requires a bandwidth for the local smoothing but also a bin size for the underlying histogram. However, McCrary [24] demonstrates that the estimation of $\hat{\theta}$ is generally robust to the choice of the bin size. We therefore refrain from reporting results for varying bin sizes and use the default bin size determined by the algorithm.

Summarizing, the test yields an estimate $\hat{\theta}$ for the discontinuity at a specific cutoff value $c$ in the distribution of test statistics by comparing the right-hand side and left-hand side limit of the distribution in $c$. Note, that the test does not use the raw distribution but rather a smoothed distribution, thereby accounting for noise in the data. It is precisely this aspect that ensures that an observed over-representation of values at the 5 percent significance level, for example, does not simply reflect rounding imprecision at the value $z = 2$. Since $\hat{\theta}$ is approximately normal, we can test whether it significantly deviates from zero by taking the ratio of $\hat{\theta}$ and its standard error $\hat{\sigma}_{\theta}$ and testing it against $H_0$: $\theta = 0$ with a z-test. Moreover, as a useful extra, estimating $\theta$ allows us to draw conclusions about the actual size of the discontinuity in $c$. With the transformation

$$\psi = \exp\left(\hat{\theta}\right) \equiv \exp\left(\ln \hat{f}^+ - \ln \hat{f}^-\right) = \exp\left[\ln\left(\frac{\hat{f}^+}{\hat{f}^-}\right)\right] = \frac{\hat{f}^+}{\hat{f}^-} \tag{5}$$

we can express the difference in the densities just above and just below the threshold of significance as a percentage value.

# 5 Cross-sectional results on publication bias

## 5.1 Cross-sectional visual analysis of publication bias

We first analyze whether our data generally indicates publication bias, pooling all coefficients over time. We exclude one-tailed test statistics ($n = 285$) for all following analyses. The most direct analysis of publication bias inspects whether there are unnatural spikes in the distribution of test statistics at the common levels of statistical significance. Fig 3 shows the distribution of test statistics for the full sample. Panel (a) shows the raw and unweighted data. The distribution is of remarkable shape and strikingly resembles the patterns documented by Bordeur [14]. We find a bimodal distribution for which the kernel density estimate prompts local maxima around $z = 2.1$ and $z = 0.5$ and a local minimum around $z = 1.3$. The position of the second local maxima at $z = 2.1$ indicates a clustering of results that are significant with at least 5 percent.

The data shows an accumulation of significant findings and an over-representation of significant results just above the thresholds of significance (i.e. publication bias). These observations are consistent with two explanations. First, researchers often test plausible hypotheses supported by theoretical considerations that have a good chance to yield significant findings. Second, the clustering at $z = 2.1$ suggests an over-representation of significant results due to preferential selection or manipulation which corresponds with our narrower interpretation of publication bias. Even though we can not explicitly test the first explanation, we acknowledge that it is unable to fully account for the local minimum at $z = 1.2$ and the valley between $z = 1$ and $z = 1.6$. Such a valley indicates the lack of expected observations and is suggestive for publication bias [14]. One consequence of these newly formed valleys or crater is the emergence of two peaks on the left and right sides, resulting from the effect shifting towards the peaks. This phenomenon is more discussed in Bordeur (et al. [14]). Essentially, the valleys play a crucial

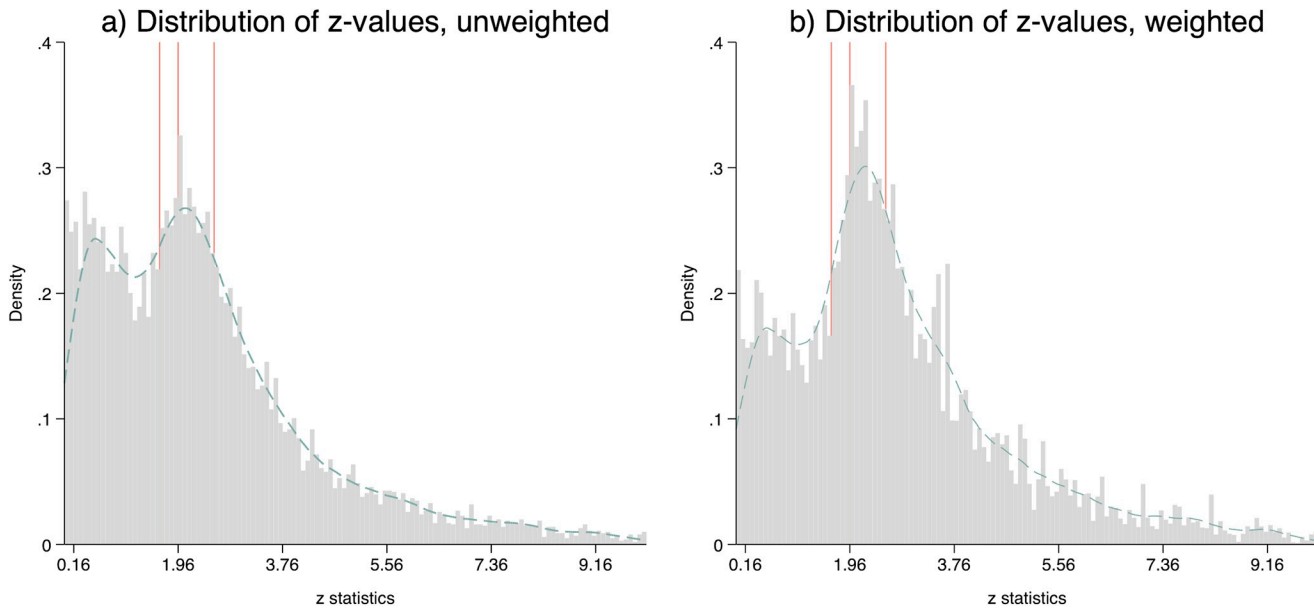

**Fig 3. Distribution of test statistics with coefficient data.** The graphs show the distribution of test statistics for the complete observation period, 1959–2018. In panel (b) the data is weighted by number of coefficients per article. The red lines indicate, in that order, the values 1.64 (10 percent significance level), 1.96 (5 percent significance level) and 2.58 (1 percent significance level).

role in selecting high-density data points, leading to the formation of local maxima. The first peak is merely displaced, implying it would not exist without the shifts in values.

Fig 3a) entails additional suggestive evidence for publication bias. Remember, that integer values might be slightly over-represented due to low rounding precision. Considering that we do not observe such notable peaks at other integer values, the prominent outlier at $z = 2$ may, however, at least to some degree indicate some form of manipulation such as favourable rounding. For illustration, consider the following example of favorable rounding. Consider a hypothetical regression coefficient of 0.585 and a corresponding standard error of 0.304. Their ratio is 1.92, yielding a test statistic which is not significant at 5 percent. However, rounding the coefficient and its standard error to two decimals, a ratio of just 1.97 is obtained, which is now *seemingly* significant at the 5 percent level. Technically, such form of rounding is mathematically correct. It nevertheless enables researchers to make results appear like being significant though they are not, without involving actual fabrication of results.

Fig 3b) shows weighted data and confirms the observations from unweighted data. Here, the test statistics are weighted by the number of coefficients per article. We again observe a bimodal distribution with a small valley in the area around $z = 1.2$ and a notable peak at $z = 2$. Hence, the overall shape is similar to panel (a) except that we observe much less non-significant results. This is consistent with the previous observation that articles that report many coefficients also report relatively more non-significant coefficients. Reducing their weight automatically results in a reduced density of coefficients in the non-significant range of the distribution.

## 5.2 Cross-sectional discontinuity analyses

In this section we present the discontinuity estimates for the full sample. As described in the method section, we apply the McCrary test to estimate the size and significance of the

discontinuity at the common thresholds of significance. The test fits two separate weighted local linear regressions to the left and the right of the specified cutoff value $c$ and compares the predicted values for $c$ estimated by the two regressions (see Eq 3). The outcome $\hat{\theta}$ is the difference of the natural logarithms of these values.

Fig 4 plots the results of the discontinuity estimations for unweighted (left column) and weighted data (right column) for the whole sample spanning the sixty years of our observation period. Panel (a) and (b) report the results for the 10 percent significance level (cutoff $c = 1.64$), panel (c) and (d) report the results for the 5 percent significance level (cutoff $c = 1.96$) and panel (e) and (f) report the results for the 1 percent significance level (cutoff $c = 2.58$). Throughout the article we report the cutoff values rounded to two decimals. For our analyses we used more precise values: $c = 1.64485$ for the 10 percent significance level, $c = 1.95996$ for the 5 percent significance level and $c = 2.57583$ for the 1 percent significance level.

There is a large discontinuity at the 10 percent threshold of statistical significance. This is clearly demonstrated in panel (a). The size of the discontinuity $\hat{\theta}$ is precisely estimated as 0.236 (*s.e.* = 0.059, $p < 0.001$). This means that the density is around 27% larger at the right-hand side of the threshold of significance than on the left-hand side ($\exp(0.236) = 1.27$). This shows a clear over-representation of results just above the 10 percent significance level. Further, there is a medium discontinuity at the 5 percent significance level (Panel (c)). However, the bias is relatively smaller compared to the 10 percent level. The discontinuity estimate is $\hat{\theta} = 0.123$, statistically significant (*s.e.* = 0.050, $p = 0.014$) and translates to an over-representation of about 13% of values just below the 5 percent significance level. Apparently, the estimation is unaffected by the peak at $z = 2$. In contrast, there is no indication of publication bias at the 1 percent level. Panel (e) shows no sign of a discontinuity at the threshold. The discontinuity estimate at the 1 percent threshold is $\hat{\theta} = -0.093$ with a standard error of *s.e.* = 0.059 and a p-value of $p = 0.116$.

We can confirm these results by analyses with weighted data. As before we weight by the number of coefficients that we extracted from the same paper. The results are shown in panel (b), (d) and (e) of Fig 4. The discontinuity estimates are 0.313 (*s.e.* = 0.064, $p < 0.001$) for the 10 percent significance level, 0.274 (*s.e.* = 0.047, $p < 0.001$) for the 5 percent significance level, and -0.025 (*s.e.* = 0.053, $p = 0.631$) for the 1 percent significance level. Overall, the estimates are notably larger for weighted data than for unweighted data. That is again the result of the fact, that papers with fewer coefficients report more significant coefficients on average and that, by weighting the data, we give more weight to those articles. However, the estimates for unweighted and weighted data are qualitatively comparable: the discontinuity is large to moderate and significant for the 10 and 5 percent significance level and small and insignificant for the 1 percent significance level.

To sum up, our cross-sectional analyses of publication bias reveals the following patterns. We find a notable and robust over-representation of test statistics just above the 10 percent significance level that can hardly be explained by rounding or testing of overly plausible hypotheses. It can, though, be attributed to preferential selection of significant results by the journal as well as the authors—in terms of not submitting non-significant results—or the pushing of results just above the threshold of significance. In addition, there is publication bias at the 5 percent significance level; however, less pronounced compared to the 10 percent level. There is no indication of publication bias at the 1 percent level.

## 5.3 Sensitivity of cross-sectional discontinuity analyses

As described before, choosing an appropriate bandwidth $h$ for estimating the discontinuities can affect the results. All graphs in Fig 4 use the default bandwidth automatically determined

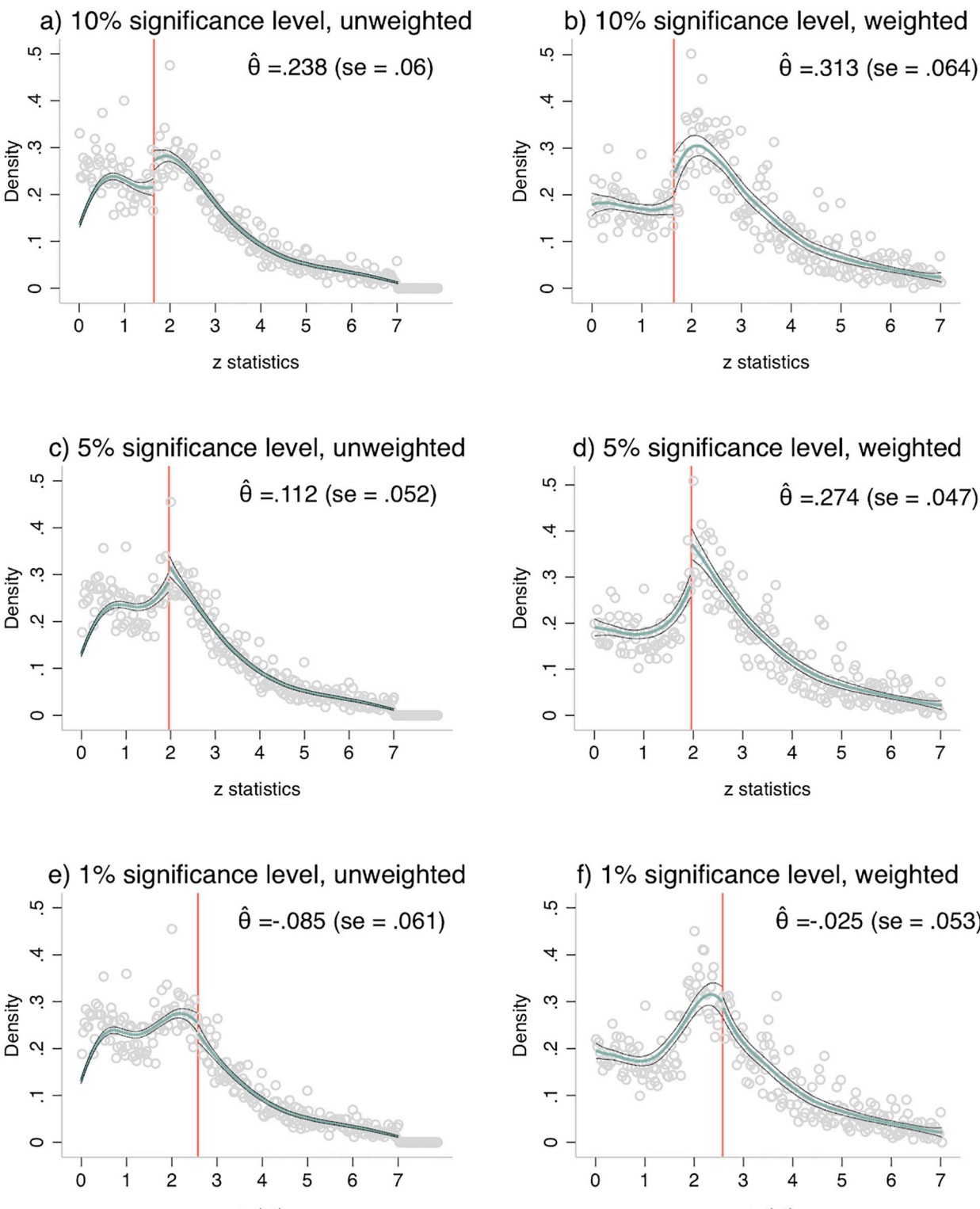

**Fig 4. Discontinuity plots, 1959–2018.** The graphs show the results of the discontinuity estimation by applying the McCrary algorithm for the 10, 5 and 1 percent significance level for unweighted, left column, and weighted data, right column, respectively. The graphs plot the distribution (grey circles) and the local linear density estimation (emerald line) with the respective 95% confidence band. Standard errors for $\hat{\theta}$ are reported in parentheses. The estimations use the respective default bandwidths and correspond to the first row, column (1), (5) and (9), in Table 2 for unweighted data and in S2 Table in S1 File for weighted data.

**Table 2. Discontinuity estimates for the common significance levels, full sample and sub-samples.**

| chosen bandwidth h | c = 1.64 | | | | c = 1.96 | | | | c = 2.58 | | | | N |
|---|---|---|---|---|---|---|---|---|---|---|---|---|---|
| | (1) | (2) | (3) | (4) | (5) | (6) | (7) | (8) | (9) | (10) | (11) | (12) | coefficients articles |
| | [default] | 0.8 | 1.2 | 1.6 | [default] | 0.8 | 1.2 | 1.6 | [default] | 0.8 | 1.2 | 1.6 | |
| 1959–2018 | 0.236*** | 0.212*** | 0.305*** | 0.372*** | 0.123** | 0.096* | 0.153*** | 0.210*** | -0.093 | -0.082 | -0.166*** | -0.233*** | 12340 |
| | (0.059) | (0.063) | (0.051) | (0.044) | (0.050) | (0.056) | (0.046) | (0.041) | (0.059) | (0.063) | (0.051) | (0.044) | 571 |
| estimated bandwidth | [0.90] | | | | [1.01] | | | | [0.89] | | | | |
| q1: 1959–1973 | 0.843** | 0.254 | 0.820** | 0.988*** | 0.206 | 0.141 | 0.228 | 0.431* | 0.897** | 0.949** | 0.370 | 0.064 | 270 |
| | (0.400) | (0.480) | (0.406) | (0.358) | (0.293) | (0.327) | (0.286) | (0.262) | (0.422) | (0.447) | (0.318) | (0.265) | 28 |
| estimated bandwidth | [1.25] | | | | [1.11] | | | | [0.86] | | | | |
| q2: 1974–1988 | 0.631*** | 0.775*** | 0.691*** | 0.590*** | 0.096 | -0.061 | 0.040 | 0.106 | -0.278 | -0.109 | -0.209 | -0.283 | 683 |
| | (0.210) | (0.289) | (0.232) | (0.193) | (0.187) | (0.239) | (0.202) | (0.178) | (0.201) | (0.274) | (0.227) | (0.194) | 51 |
| estimated bandwidth | [1.40] | | | | [1.43] | | | | [1.50] | | | | |
| q3: 1989–2003 | 0.079 | 0.080 | 0.118 | 0.187** | 0.050 | 0.002 | 0.066 | 0.136* | -0.174* | -0.177 | -0.185** | -0.204** | 3876 |
| | (0.108) | (0.113) | (0.093) | (0.080) | (0.092) | (0.108) | (0.088) | (0.077) | (0.094) | (0.110) | (0.091) | (0.079) | 198 |
| estimated bandwidth | [0.88] | | | | [1.09] | | | | [1.11] | | | | |
| q4: 2004–2018 | 0.266*** | 0.220*** | 0.343*** | 0.421*** | 0.159** | 0.153** | 0.200*** | 0.248*** | -0.088 | -0.048 | -0.164** | -0.249*** | 7511 |
| | (0.074) | (0.080) | (0.065) | (0.056) | (0.064) | (0.070) | (0.058) | (0.051) | (0.074) | (0.081) | (0.065) | (0.055) | 294 |
| estimated bandwidth | [0.93] | | | | [0.95] | | | | [0.96] | | | | |

Note: Each cell reports the results of a single McCrary discontinuity estimation $\hat{\theta}$. In column (1), (5) and (9) we report in brackets the bandwidth that the local linear smoothing procedure determined by default. Additionally, we vary the bandwidth $h$ in the other columns to examine the robustness of our results. We perform the analyses for the 10 percent significance level, indicated by $c = 1.64$, in columns (1)-(4), for the 5 percent significance level, indicated by $c = 1.96$, in columns (5)-(8) and for the 1 percent significance level, indicated by $c = 2.58$, in columns (9)-(12) Standard errors are reported in parentheses.

\* $p < 0.10$,

\*\* $p < 0.05$,

\*\*\* $p < 0.01$.

by the estimation algorithm. To test the sensitivity of our results to the choice of the bandwidth, we repeat the analyses for varying bandwidths and report the results in Table 2, first row (we provide the according results for weighted data in S2 Table in the S1 File).

Columns (1)-(4) refer to the 10 percent significance level. In column (1) we reproduce the results from the discontinuity plot. The following columns show the result for the bandwidths 0.8, 1.2 and 1.6 (see S3 Fig in S1 File for the corresponding discontinuity plots). The sensitivity results at the 10 percent significance threshold confirm our main analyses. There is a slight increase of $\hat{\theta}$ with increasing bandwidth $h$. We further see a highly significant discontinuity at the 10 percent significance level for all choices of $h$, ranging from 0.21 to 0.37. Inspection of the discontinuity plots suggests that the differences between the estimates result from the right-hand side of the local smoothing. The close-by local maximum naturally has a stronger effect on the discontinuity estimation at $c = 1.64$ for larger bandwidths. This also shows a general difficulty with separating analyses of the different significance levels with large bandwidths. Since the cutoff values $c$, for which we estimate the discontinuity, lie close to each other, discontinuities at a particular point may affect the estimation at other points. For instance, the discontinuity estimate for $c = 1.64$ may be inflated by the peak at $z = 2$, and, vice

versa, the estimate for $c = 1.96$ may be deflated by the discontinuity at $c = 1.64$. However, this can be circumvented by choosing smaller values for the bandwidth $h$ and critical inspection of the corresponding discontinuity plots.

The sensitivity analyses at the 5 and the 1 percent significance level also confirm our main results. Table 2, columns (5)-(12), first row, show the respective results (further, see also S4 and S5 Figs in the S1 File). The estimates for the 5 percent level are about half the size compared to the estimates for the 10 percent level and at least marginally significant for all bandwidths. The discontinuity plots show a better fit for smaller bandwidths and the peak at $z = 2$ does not seem to substantially affect the estimation. As in our main analyses, we observe no discontinuity for the 1 percent significance level. Either our estimation procedure is unable to detect a discontinuity or there is no discontinuity. Inspection of the discontinuity plots suggests that both explanations may be valid. As for the 10 percent significance level, the close-by local maximum substantially biases the estimation particularly for larger bandwidths. Since the local maximum is now at the left-hand side of cutoff, we even observe negative discontinuity estimates. Relying on smaller bandwidths, we see that there is no discontinuity at $c = 2.58$.

Overall, the sensitivity analyses confirm our main results. We find a notable and robust over-representation of test statistics just above the 10 percent significance level. There is a moderate over-representation of significant results at the 5 percent level. There is no indication of publication bias at the 1 percent level.

## 6 Longitudinal results on publication bias

Since the main objective of our study is to investigate whether patterns of publication bias have changed over time, we extend the analyses from the previous section and take account for the longitudinal nature of our data. For a first inspection, we present results for four separate observational periods. Therefore, we divide the 60 years into 15-year intervals. Table 2, row (q1) to (q4), reports the discontinuity estimates for the four sub-samples for the 10, 5 and 1 percent significance level (the corresponding discontinuity plots for each estimate are provided in the S3 to S17 Figs in S1 File).

Our results demonstrate a shift from publication bias at the 10 percent level to publication bias both at the 10 percent and at the 5 percent level. Columns (1) to (4) provide results for the 10 percent significance level. The discontinuity is large and significant for the first half of our observation period. With respect to the default bandwidth, the estimates range between 0.63 and 0.84. For the second half of our observation period, in contrast, we observe much smaller discontinuity estimates that are only significant for the most recent period. Overall, this indicates that the bias at the 10 percent significance level was substantial in the sixties through eighties of the last century, decreased in the nineties and increased again in recent years although the effect raised to a much lower level than in the first half of the observation period. In column (5) to (8) we report the results for the 5 percent significance level. In contrast to the 10 percent significance level, the discontinuity is largest for the most recent years only. Though substantial, the bias of 0.16 for the 5 percent level in the most recent period is still lower than the bias of 0.27 for the 10 percent level. There is no substantial discontinuity and no significant trend of publication bias at the 1 percent level. Column (9) to (12) report the results for the 1 percent significance level. The discontinuity is moderate and only marginally significant for the first quarter of our observation period. For the second to fourth quarter, in contrast, the estimated discontinuity is small and negative. As discussed in the previous section we are less confident on whether we are able to reliably estimate the discontinuity at the 1 percent significance level.

To fully exploit the longitudinal nature of our data, we generalize our analyses by what we call *moving discontinuities*. This analysis considers varying interval sizes and boundaries. This is particularly useful considering that we extracted considerably less coefficients in earlier years than in recent years. Further, this ensures that our results do not depend on a particular choice of sub-samples.

The details of the procedure are as follows. We compute *moving discontinuities* comparable to the principle of moving averages:

$$
\begin{aligned}
\hat{\theta}_{1959,k} &= \hat{\theta}(Z_{1959}, Z_{1960}, \ldots, Z_{1959+k-1}) \\
\hat{\theta}_{1960,k} &= \hat{\theta}(Z_{1960}, Z_{1961}, \ldots, Z_{1960+k-1}) \\
&\vdots \\
\hat{\theta}_{2018-k+1,k} &= \hat{\theta}(Z_{2018-k+1}, Z_{2018-k+2}, \ldots, Z_{2018})
\end{aligned}
\tag{6}
$$

where $Z_y$ is the set of test statistics extracted in year $y$. For given interval length $k$ we, first, estimate the discontinuity $\hat{\theta}_{1959,k}$ for the first $k$ years, 1959 to $1959 + k - 1$. We then *move* the interval by one year and calculate the discontinuity $\hat{\theta}_{1960,k}$ for the next $k$ years starting with 1960. Continuing this procedure, we end with $60 - k + 1$ discontinuity estimates for $60 - k + 1$ corresponding intervals whereas neighboring intervals always overlap by $k - 1$ years. This procedure, as in moving averages, clearly introduces strong correlation between successive waves (approximately 14/15 of the observations are the same in both waves). As we will explain below, we do not make significance claims for these moving discontinuities and therefor do not consider corrections for auto-correlation, nor do we explicitly model it. Note that we use the default bandwidth determined by the estimation procedure for all analyses since the results and discontinuity plots reported in the previous section suggest that it generally yields reliable estimates. With this procedure we make sure that our results do not depend on a particular division of our observation period. We further vary the interval length $k$ to check for robustness. This means that for larger $k$ we will have higher power and more precise discontinuity estimates since the single intervals contain more values. On the other hand we smooth over possible temporal variations. In contrast, smaller $k$ will result in a more fine-grained evaluation of temporal changes, but yield less precise estimates. To navigate between the pros and cons of smaller and larger $k$, we computed *moving discontinuities* for $k = 5, \ldots, 20$. Since the detected patterns are comparable, we only report the results for $k = \{10, 15, 20\}$. Note also, that our suggested approach to investigate time trends of publication bias does not depend on the discontinuity method, but can be applied to any publication detection method.

We report our key results for $k = 15$ in Fig 5. All graphs are computed using confidence bands for the 95% confidence level. The patterns for moving discontinuities strikingly confirm the results from discrete analyses, demonstrating a trend from publication bias at the 10 to the 5 percent level. Panel (a), (b) and (c) of Fig 5 show the results for the 10 percent, 5 percent and 1 percent significance level for $k = 15$. The results, therefore, directly correspond to Table 2.

Panel (a) demonstrates a decrease of publication bias at the 10 percent level. The discontinuity is very high in the early years of our observation period with a $\hat{\theta}$ ranging steadily around 1. In the first half of the observation period the discontinuity is highly significant for all intervals indicating a large over-representation of results just above the 10 percent significance level. The period from 1978 to 1992 records a notable drop and is the first interval for which the discontinuity is not significant anymore. The discontinuity remains small and significant, with values close to zero, until shortly before 2010. After that it slowly increases to a stable and significant discontinuity around $\hat{\theta} = 0.3$.

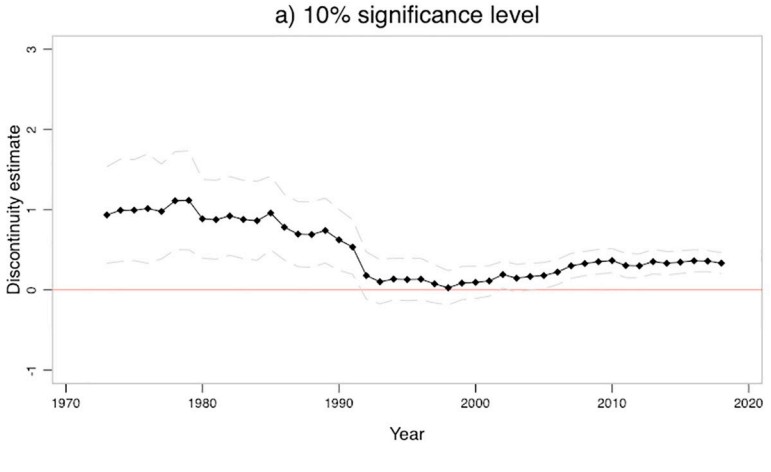

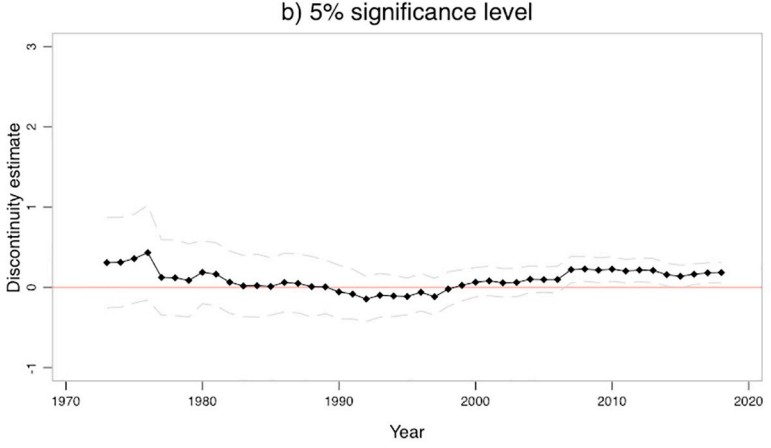

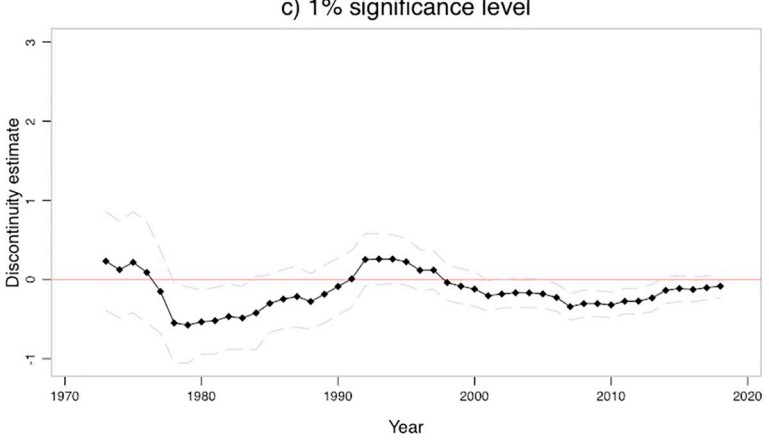

**Fig 5. Moving discontinuity plots.** The graphs plot *moving discontinuity* estimates for the 10, 5 and 1 percent significance level for $k = 15$ *years* time windows. Each dot is the result of a single discontinuity estimation. The x-axis marks the end point of the respective intervals. For example, the first dot reports the estimate for the interval 1959–1973, the second dot reports the estimate for the interval 1960–1974, and so forth. The outer lines denote the 95% confidence interval. The estimates are reported in Table 2.

Panel (b) provides the graph for the 5 percent significance level. The discontinuity is insignificant and small for much of the earlier observation period. Then, there is an increase in publication bias after 2007 and the discontinuity estimates become significant from then on, ranging between 0.1 and 0.2.

Panel (c) shows discontinuity trends for the 1 percent significance level. There is no substantial trend and most of the time, discontinuities are insignificant. More in particular, the discontinuity estimates start large but insignificant and quickly drop into the negative range in the eighties. Even though we see some fluctuations after that, the discontinuity remains negative for most of the intervals. Around 2010 the estimates become significant, but for reasons already discussed, we do not suggest to over-interpret this finding.

We have run a number of sensitivity analyses, confirming our main findings. In particular, we have varied the interval lengths *k* and repeated the analyses with the caliper test. S18 and S19 Figs in the S1 File report intervals of *k* = 10 and *k* = 20. As expected, the plots are more volatile and have wider confidence bands for 10 year *moving discontinuities*. In contrast, the plots reporting 20 year *moving discontinuities* are very smooth, but have narrower confidence bands. Both variations replicate the shift from the 10 to the 5 percent level.

## 7 Discussion

This study contributes to the increasing body of literature that investigates whether and how publication bias has changed over the last decades. Drawing on a sample of more than 600 empirical papers published in the *Quarterly Journal of Economics* during the last six decades, we test for discontinuities at the common levels of significance and identify several clear patterns.*First*, we observe a significant publication bias over the last six decades. Our data indicates a substantial over-representation of statistically significant effects just above the common thresholds of significance in the observed period. We have developed a new method by applying the McCrary test procedure known from regression discontinuity to detect discontinuities at the common thresholds of significance and our findings are robust across several parameter variations.

*Second*, we observe a substantial change in bias patterns over time. Our observation period can roughly be devided into three phases. In the first phase, spanning the first two quarters, we find a large and stable bias at the 10 percent significance level, indicating the importance of that specific threshold in those years. Around the year 1990, we observe a transition into the second phase, which roughly comprises the third quarter of our observation period. In these years we do not observe any publication bias, neither at the 10 nor at the 5 percent level of significance. Lastly, the third phase spans the most recent quarter of our observation period and shows substantial and robust bias as well at the 10 and the 5 percent level of significance. This implies that publication bias moved to some degree from the 10 to the 5 percent level, indicating that the more rigorous threshold of 5 percent became increasingly important for researchers to push their results beyond and for reviewers as a selection criterion of what to recommend for publication and what to reject.

How can these remarkable changes be explained? We first note that the large bias at the 10 percent threshold before 1990 coincides with the low proportion of empirical papers among the published articles. Before 1990, less than 15% of the published articles contained an empirical study. Even though one could argue that only few empirical studies have been conducted at that time, the results on publication bias might suggest a different story. Finding large discontinuities –an indicator of publication bias– before 1990 can likewise indicate that empirical papers have faced comparably high publication barriers represented by a low publication rate of such papers.

This argumentation is further supported by the absence of bias in the years after 1990. The substantial increase in the proportion of empirical papers after 1990 presumably indicates a higher acceptance rate for such papers, and, therefore, a temporarily reduced competition. A *Manifesto* published by the editors of the *QJE* in which they announce a reorientation of the journal also suggests that the transition towards more empirical papers reflects an editorial decision on what kind of papers are selected for publication, rather than a change in what kind of papers are submitted for publication by researchers [49]. The large bias before 1990 and the absence of any bias thereafter is therefore consistent with the argument that this is the result of changes in the publication chance of empirical papers.

The second noteworthy observation is the shift from a publication bias only at the 10 percent significance level in earlier decades to a publication bias at both the 10 and 5 percent level in recent years. From a statistical point of view, the explanation can be found in the sample sizes and degrees of freedom, which have increased sharply over time. The *p* values are sensitive to changes in sample size, especially for smaller sample sizes, and for a fixed test statistic, the *p* value decreases quasi-exponentially with increasing sample size. Considering that 75% of the coefficients that we extracted from articles *before* 1990 are linked to a sample size below 350 and that, in contrast, 75% of the coefficients that we extracted *after* 1990 are linked to a sample size above 368, we believe that the 10 percent significance level may have been more attractive to researchers before 1990 simply because sample sizes and, therefore, statistical power were notably smaller at that time. In other words, the size of the bull's-eye was adjusted to the changing circumstances in such a way that there was a better chance of getting a hit in the form of a significant result. This could also explain that the proportion of significant results remained constant despite increasing sample size.

Lastly, we discuss the overall impact that publication bias may have or has had on economics and related social sciences today and in the past. We make two arguments that suggest that the absolute magnitude and influence of biased research has increased. First, even though publication bias was apparently already present and large in earlier decades, it played a minor role since it only affected a small share of economic research. In contrast, the majority of articles in recent decades were of empirical nature. Therefore, although the relative magnitude of the bias may have become smaller, the overall spread of biased research has become larger and, therefore, it became a more wide-spread problem than it used to be.

Our *second* argument goes beyond our findings and addresses general trends that can be observed in highly influential journals. The *QJE* belongs to the elite group of highly regarded journals and the constantly increasing number of submissions has pushed their acceptance rate into the single-digit range (see for example [38], and our own analysis in Section 3 in the S1 File), simultaneously increasing the pressure on and the competition between researchers that aim for publications in one of these journals. In light of the above discussion, a lower acceptance rate will go hand in hand with greater selectivity in choosing which papers to publish. These highly influential journals and their published contents, therefore, become more and more relevant for the career of junior researchers. For example, an increasing number of tenure requirements at universities demand publications in such top journals. These regulations are highly controversial. Heckman and Moktan [50] dubbed the competition for the positional goods of rare places in the five leading journals, the Quarterly Journal of Economics among them, as the "tyranny of the big five." However, their leading position makes articles in top journals idealized role models of how and what to publish. As a consequence, more and more researchers relate to the "biased" literature in these journals when selecting research topics and relating new research to the current state of the art. In this sense, publication bias in top journals can have reinforcing dynamics, influencing a whole series of upcoming publications and research trends based on innovative, but presumably biased studies.

## 8 Conclusion

To summarize, our findings draw a picture that shows an increasing trend from theory to empirics, towards larger studies with more complex statistics and towards more rigorous criteria for statistical significance testing. This increasing competition and demand for elaborate empirics may have led researchers nowadays to push their analyses beyond the 5 percent significance level, while in earlier times, pushing beyond the 10 percent level may have been regarded as enough.

Examining how publication bias has evolved and changed in the past raises several important conclusions about how publication bias might change in the future. We, therefore, conclude by discussing several policy implications of what can be done to reduce publication bias in the future. There have been a number of suggestions from editors, authors and, more generally, from advocates of the open science movement. They range from interventions (1) during the publication process by implementing more rigorous statistical conventions and changing the review process, (2) after the publication process by creating stronger incentives for replications, and (3) before the publication process by demanding pre-registration.

One reaction to the increasing evidence on bias in empirical research is to demand more rigorous thresholds of statistical significance. The *American Sociological Review*, one of the top journals in sociology, recommends not to use the 10 percent significance level any more [51], others in interdisciplinary forums (*Nature Human Behavior*) go even further calling for significance levels far below 5 percent [52]. However, the problem may merely be shifted once stricter criteria have been implemented. Our analysis shows that publication bias in economics shifted from biased research at the 10 percent level towards biased research at the 5 percent level during the times when criteria became stricter. If the community asks now for stricter significance levels and they became standard, it is plausible that future biased research will occur at the new levels. Although one may argue that it becomes increasingly harder to push evidence above stricter criteria, stricter significance criteria also require larger samples for all researchers, relativizing the required efforts to produce less biased research once again. Instead of evermore stricter significance criteria, reporting effect sizes along with the statistical results may actually help shifting the focus from significance to meaningfulness [53].

As an alternative to suggesting stricter criteria for performing statistics, it has been suggested to change the review process. In order to overcome publication bias, a results-free review process has been put forward and initial trials have already been conducted by several journals [54, 55]. The idea is that the evaluation of manuscripts should be independent of the reported statistical results. This may be one promising way to foster the publication of manuscripts independently from significance levels, *p* values and the like, since it separates the review process from the outcomes of the study. However, first implementations of this approach yielded mixed results. Separating the conceptual part of manuscripts from their statistical results raises new problems such as a shift in incentives and increased costs, e.g. reduced transparency since information about data and results is withheld from editors and reviewers, or a flood of null findings since researchers may be encouraged to open their "file-drawers" [54, 56].

Changes after the publication process have also been suggested. In particular, creating stronger incentives for replications may change what will be published in the future since they enhance the likelihood of error detection. Consequently, false-positive and inflated estimates might undergo a correction process in the literature. Moreover, if researchers anticipate replications to come, they may be more cautious or reluctant to publish "statistical outliers" and biased results. In this way, stronger incentives for conducting and publishing replication studies may reduce biased research. However, it is well known that replications are to be

understood as a contribution to a collective good. Therefore, "selective incentives" [57] are needed so that the scientific community can benefit from replications. In particular, sensitizing junior scientists for publication bias and creating respective incentives, such as asking for replication studies already in Master or Ph.D. programs, could further contribute to less bias in future published knowledge (see for example the recommendations by Diekmann [58] for economics programs and the suggestions by King [59, 60] for political science programs at Harvard). In addition, it has been recommended to change journal data policies towards demanding open data sharing. If this policy became standard, the costs for performing replications would be dramatically reduced, researchers would anticipate replications and may be less inclined to publish biased results.

In addition to changing the process after data collection (statistics, reviews and replications), a change of the policies before data collection may even be more effective. In particular, pre-registration could be a promising mechanism. In the meanwhile, a number of public data repositories have been installed and a rise in pre-registration of studies can be observed. For clinical trials in the US a pre-registration of studies is even required by law [61]. Pre-registration increases the incentives to publish null-findings and decreases possibilities to select research findings based on their significance. A recent study suggests that pre-registration may actually reduce publication bias and promote the publication of null findings [62]. The trend towards pre-registration is aligned with upcoming new journals specializing on publishing studies for which the data refuted prior hypotheses and expectations (e.g. the *Journal of Negative Results*). This certainly decreases the costs for publishing pre-registered (and other kinds of) null findings and increases the incentives for pre-registration.

All in all, reflecting on our evidence in tandem with our discussion of policy interventions in the scientific publication process, we conclude that the transition towards less biased research is a long way to go. A number of measures have been suggested, tested and implemented. Nevertheless, their effectiveness and impact and their possibly negative side-effects and counter-intuitive macro-transitions will have to be empirically evaluated in future studies.

## Supporting information

**S1 File.**
(PDF)

## Acknowledgments

For helpful comments, we are grateful to participants of the MAER-Net Colloquium at Deakin University, Workshop on Analytical Sociology at Venice International University, the PRINTEGER European Conference on Research Integrity at Bonn University, the ESRA conference at ISEG Lisbon School of Economics Management and seminar participants at University of Zurich, LMU Munich and ETH Zurich. We thank Alexander Ehlert and Andreas Diekmann for outstanding research assistance. We particularly thank Alexander Ehlert for his support in adapting the R-command for the McCrary test. Lastly, we want to thank the editor and the anonymous reviewers for helpful and constructive feedback which has been highly valuable for improving the paper.

## Author Contributions

**Conceptualization:** Julia Jerke.

**Data curation:** Julia Jerke, Antonia Velicu.

**Formal analysis:** Julia Jerke, Heiko Rauhut.

**Funding acquisition:** Heiko Rauhut.

**Investigation:** Julia Jerke, Antonia Velicu, Heiko Rauhut.

**Methodology:** Julia Jerke.

**Project administration:** Heiko Rauhut.

**Resources:** Julia Jerke.

**Software:** Fabian Winter.

**Supervision:** Heiko Rauhut.

**Validation:** Antonia Velicu, Fabian Winter, Heiko Rauhut.

**Visualization:** Julia Jerke, Fabian Winter.

**Writing – original draft:** Julia Jerke, Heiko Rauhut.

**Writing – review & editing:** Julia Jerke, Antonia Velicu, Fabian Winter, Heiko Rauhut.

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
