## [Decision Letter · Decision Letter 0]

20 Feb 2024

PONE-D-23-43188Publication Bias in the Social Sciences since 1959: Application of a Regression Discontinuity FrameworkPLOS ONE

Dear Dr. Rauhut,

Thank you for submitting your manuscript to PLOS ONE. After careful consideration, we feel that it has merit but does not fully meet PLOS ONE’s publication criteria as it currently stands. Therefore, we invite you to submit a revised version of the manuscript that addresses the points raised during the review process.This paper examines the progression of publication bias, particularly in relation to the use of statistical tests in research. The authors observe a trend towards larger studies that employ more sophisticated statistical analyses and adhere to stricter testing criteria. The manuscript is well-composed and utilizes suitable methodologies. Nonetheless, there are a few minor areas that require enhancement. Please review the comments with attention and respond accordingly.

Please submit your revised manuscript by Apr 05 2024 11:59PM. If you will need more time than this to complete your revisions, please reply to this message or contact the journal office at plosone@plos.org. Please include the following items when submitting your revised manuscript:A rebuttal letter that responds to each point raised by the academic editor and reviewer(s). You should upload this letter as a separate file labeled 'Response to Reviewers'.A marked-up copy of your manuscript that highlights changes made to the original version. You should upload this as a separate file labeled 'Revised Manuscript with Track Changes'.An unmarked version of your revised paper without tracked changes. You should upload this as a separate file labeled 'Manuscript'.If applicable, we recommend that you deposit your laboratory protocols in protocols.io to enhance the reproducibility of your results. Protocols.io assigns your protocol its own identifier (DOI) so that it can be cited independently in the future. For instructions see: https://journals.plos.org/plosone/s/submission-guidelines#loc-laboratory-protocols. Additionally, PLOS ONE offers an option for publishing peer-reviewed Lab Protocol articles, which describe protocols hosted on protocols.io. Read more information on sharing protocols at https://plos.org/protocols?utm_medium=editorial-email&utm_source=authorletters&utm_campaign=protocols.

We look forward to receiving your revised manuscript.

Kind regards,

Yuyan Wang, Ph.D.

Academic Editor

PLOS ONE

Journal Requirements:

"Swiss National Science Foundation; Starting Grant CONCISE, BSSGI0-155981"

5. Please amend your manuscript to include your abstract after the title page.

Reviewers' comments:

Reviewer's Responses to Questions

**Comments to the Author**

1. Is the manuscript technically sound, and do the data support the conclusions?

Reviewer #1: Yes

Reviewer #2: Yes

2. Has the statistical analysis been performed appropriately and rigorously? 

Reviewer #1: Yes

Reviewer #2: Yes

3. Have the authors made all data underlying the findings in their manuscript fully available?

Reviewer #1: Yes

Reviewer #2: Yes

4. Is the manuscript presented in an intelligible fashion and written in standard English?

Reviewer #1: Yes

Reviewer #2: Yes

5. Review Comments to the Author

Reviewer #1: Summary: This paper provides a comprehensive analysis of the evolution and current trends in publication bias, primarily in the context of statistical significance testing in empirical research. The authors highlight a shift from theory-based research to larger empirical studies with increasingly complex statistics and stringent significance testing criteria. They discuss the implications of this shift for future research and publication bias, addressing the potential policy changes that could mitigate such biases. The study employs a longitudinal approach to evaluate the changing standards over time, underscoring the movement from the acceptance of the 10 percent significance level to the current preference for the 5 percent or even lower thresholds. The paper further delves into the various interventions proposed by the open science movement to reduce publication bias, such as modifications in the peer-review process, the push for replication studies, and the advocacy for pre-registration and open data policies. The authors conclude that while steps towards reducing biased research are being taken, the journey to overcome these biases is long and requires ongoing evaluation and adaptation of scientific publication practices.

I have some queries and suggestions to improve the manuscript:

1). I would like to request further details regarding the application of the McCrary test as outlined on page 19. Specifically, the manuscript would benefit from a clearer explanation of the smoothing technique applied to the histogram of the test statistic distribution. A more comprehensive discussion on the potential effects of over-smoothing or under-smoothing on the study's findings would greatly enhance the robustness of the methodology presented

2). On page 21, regarding Equation 4, the manuscript does not clearly articulate the process for selecting the bandwidth $h$. For the sake of clarity and reproducibility, a detailed explanation of the criteria and rationale behind the choice of bandwidth would be beneficial to the reader's understanding.

3). In the presentation of the cross-sectional results, specifically observed in Figure 3, there are notable local maxima near z = 2.1 and z = 0.5. The peak at z = 0.5 is particularly intriguing and appears to be insufficiently explained within the current manuscript. An in-depth analysis of this observation and its potential impact on the study's conclusions would be of great interest and could provide valuable insights into the robustness of the findings.

4). On page 27, the authors introduce a 'moving discontinuities' approach to examine longitudinal effects within the dataset. However, the described methodology implies that each interval overlaps with its neighbor by k−1 years, potentially leading to autocorrelation in the estimated values. The treatment of this potential autocorrelation is not sufficiently detailed in the text.

5). In reference to Figure 5, The grey bands highlight the estimation results for disjoint intervals, but it is not clear how these bands are calculated or what they represent statistically. This could be a point of confusion for readers.

6). For Table 2 and Table A3, it would enhance clarity to reposition the bandwidth information currently placed at the bottom of the table. Additionally, a brief explanatory note on the significance of the numerical values presented within parentheses "()" and brackets "[]" would contribute to a more intuitive interpretation of the table's data.

7). Figures A1 and A2 are quite similar. It would be beneficial to provide some markers directly on the plots.

Reviewer #2: The authors proposed a study of publication bias spanning 60 years using a regression discontinuity design. The manuscript is well-organized and written clearly. The research question is presented with commendable clarity, and the chosen study design logically aligns with the research objectives. The discussions are closely tied to the analysis results and are concise. However, one minor comment is that the manuscript contains very detailed explanations for nearly every concept and method, in both social science and statistical science, which makes it verbose and excessively long. The authors might consider revising it to make it more concise and reader friendly.

6. PLOS authors have the option to publish the peer review history of their article (what does this mean?). If published, this will include your full peer review and any attached files.

Reviewer #1: No

Reviewer #2: No

---

## [Decision Letter · Decision Letter 1]

4 Jun 2024

Publication Bias in the Social Sciences since 1959: Application of a Regression Discontinuity Framework

PONE-D-23-43188R1

Dear Dr. Rauhut,

We’re pleased to inform you that your manuscript has been judged scientifically suitable for publication and will be formally accepted for publication once it meets all outstanding technical requirements.

Kind regards,

Yuyan Wang, Ph.D.

Academic Editor

PLOS ONE

Additional Editor Comments (optional):

Reviewers' comments:

Reviewer's Responses to Questions

**Comments to the Author**

1. If the authors have adequately addressed your comments raised in a previous round of review and you feel that this manuscript is now acceptable for publication, you may indicate that here to bypass the “Comments to the Author” section, enter your conflict of interest statement in the “Confidential to Editor” section, and submit your "Accept" recommendation.

Reviewer #1: All comments have been addressed

2. Is the manuscript technically sound, and do the data support the conclusions?

Reviewer #1: Yes

3. Has the statistical analysis been performed appropriately and rigorously? 

Reviewer #1: Yes

4. Have the authors made all data underlying the findings in their manuscript fully available?

Reviewer #1: Yes

5. Is the manuscript presented in an intelligible fashion and written in standard English?

Reviewer #1: Yes

6. Review Comments to the Author

Reviewer #1: The authors have addressed the reviewers' comments satisfactorily, and the paper is now suitable for acceptance.

7. PLOS authors have the option to publish the peer review history of their article (what does this mean?). If published, this will include your full peer review and any attached files.

Reviewer #1: No

---

## [Editor Report · Acceptance letter]

17 Jul 2024

PONE-D-23-43188R1 

PLOS ONE

Dear Dr. Rauhut, 

I'm pleased to inform you that your manuscript has been deemed suitable for publication in PLOS ONE. Congratulations! Your manuscript is now being handed over to our production team.

Kind regards, 

on behalf of

Dr. Yuyan Wang 

Academic Editor

PLOS ONE